# Genome-Wide Identification and Molecular Evolutionary History of the Whirly Family Genes in *Brassica napus*

**DOI:** 10.3390/plants13162243

**Published:** 2024-08-13

**Authors:** Long Wang, Zhi Zhao, Huaxin Li, Damei Pei, Qianru Ma, Zhen Huang, Hongyan Wang, Lu Xiao

**Affiliations:** 1Academy of Agricultural and Forestry Sciences, Qinghai University, Xining 810016, China; 2Laboratory for Research and Utilization of Qinghai Tibet Plateau Germplasm Resources, Xining 810016, China; 3Key Laboratory of Spring Rapeseed Genetic Improvement of Qinghai Province, Xining 810016, China; 4Qinghai Spring Rape Engineering Research Center, Xining 810016, China; 5State Key Laboratory of Crop Stress Biology for Arid Areas, College of Agronomy, Northwest A&F University, Yangling 712100, China; 6Laboratory of Plant Epigenetics and Evolution, School of Life Science, Liaoning University, Shenyang 110036, China

**Keywords:** *Brassica napus* L., Whirly, evolutionary analysis, expression pattern, abiotic stress

## Abstract

Whirly transcription factors are unique to plants, playing pivotal roles in managing leaf senescence and DNA repair. While present in various species, their identification in *Brassica napus* L. (*B*. *napus*) and their differences during hybridization and polyploidy has been elusive. Addressing this, our study delves into the functional and evolutionary aspects of the Whirly gene family during the emergence of *B. napus*, applying bioinformatics and comparative genomics. We identified six *Whirly* genes in *B*. *napus.* In *Brassica rapa* L. (*B. rapa*), three *Whirly* genes were identified, while four were found in *Brassica oleracea* L. (*B. oleracea*). The results show that the identified *Whirly* genes not only have homology but also share the same chromosomal positions. Phylogenetic analysis indicates that *Whirly* genes in monocots and dicots exhibit high conservation. In the evolutionary process, the Whirly gene family in *B*. *napus* experienced events of intron/exon loss. Collinearity insights point to intense purifying selection post-duplication. Promoter regions housed diverse cis-acting elements linked to photoresponse, anaerobic initiation, and methyl jasmonate responsiveness. Notably, elements tied to abscisic acid signaling and meristem expression were prominent in diploid ancestors but subdued in tetraploid *B*. *napus*. Tissue-specific expression unveiled analogous patterns within subfamily genes. Subsequent qRT-PCR analysis spotlighted *BnAWHY1b*’s potential significance in abiotic stress response, particularly drought. These findings can be used as theoretical foundations to understand the functions and effects of the Whirly gene family in *B*. *napus*, providing references for the molecular mechanism of gene evolution between this species and its diploid ancestors.

## 1. Introduction

Polyploidization, a common event in the diversification and evolution of plants, holds considerable importance for angiosperm speciation [1,2]. This process is often accompanied by genome rearrangement, homologous gene loss, and alterations in gene expression patterns [3]. Concurrently, changes in gene families, such as expansion, may occur in plants [4]. Transcription factors are crucial for plant growth and development, serving key roles in various physiological processes. These proteins have been shown to be integral in responding to both biotic and abiotic stresses, and they are also pivotal in mediating plant hormone signal transduction [5,6,7,8]. Whirly proteins constitute a small family of plant-specific transcription factors primarily found in chloroplasts and plastids, with a few localized in mitochondria [9,10,11]. These proteins consist of three distinct functional domains: the N-terminal domain, which typically contains signal peptides for chloroplasts or mitochondria; the Whirly domain, responsible for binding single-stranded DNA; and the C-terminal variable domain, which regulates this binding activity [12]. Studies reveal that Whirly transcription factors play crucial roles in regulating leaf senescence [13,14], silique development [15,16], plastid and mitochondrial stabilization, and genomic damage repair [17,18,19,20,21]. Moreover, maintaining stable genetic information expression is vital for plant growth and development. Consequently, these multifunctional transcription factors have been extensively studied in various plants, including *Arabidopsis thaliana* L. [9,19], *Oryza sativa* L. [22,23], *Zea mays* L. [15,17], *Solanum tuberosum* L. [24], and *Medicago sativa* L. [25].

Whirly transcription factors were first identified in potatoes. PBF-2 (PR-10a binding factor 2) can combine with the cis-acting element ERE on the promoter of the pathogenesis-related gene (PR-10a) to regulate the expression of defense genes [24]. Three *Whirly* genes (*AtWHY1-3*) were identified in *A*. *thaliana* by Krause et al., of which *AtWHY1* and *AtWHY3* were mainly localized in chloroplasts or plastids, while *AtWHY2* was localized in mitochondria [9]. The *WHY1* gene plays a dominant role in regulating the leaf senescence and repairing DNA damage. For example, the content of H_2_O_2_ is remarkably increased when *WHY1* is knocked out from the seeds of *A*. *thaliana*, eventually resulting in the senescent phenotype in plants [26]. In addition, the deletion of *WHY1/3* significantly inhibits the binding between RNA polymerase and DNA, which diminishes the homologous recombination repair [27]. *WHY1* is also actively involved in the signal transduction of plant hormones, such as salicylic acids and abscisic acids [16,28]. Besides the important effects on the growth and development of plants, this gene can also respond to abiotic stresses. Bonding with the promoter region, *SlWHY1* activates the expression of *SlRbcS1*, making it respond to low-temperature stresses [29]. Furthermore, *SlWHY1* also serves as a heat stress-induced gene to increase the heat resistance of tomatoes by regulating the expression of *SlHSP21.5A* [30]. *WHY1* and *WHY3* can interact with each other to maintain the stability of genomes [20,31]. *WHY2* actively participates in the signal pathway of methyl jasmonate. It is of great significance to repair and stabilize the mitochondrial genomes when acting as a binding protein of DNA/RNA in mitochondria, despite its undefined mechanism of action in chloroplasts or mitochondria [32].

*Brassica napus* L. (*B*. *napus*), a characteristic allotetraploid species within the *Brassica* genus, arises from the hybridization and polyploidization of *B*. *rapa* and *B*. *oleracea*, and it holds significance as a major oil crop. The Whirly gene family in *B*. *napus* has not yet been thoroughly examined regarding its number, characteristics, and roles. Consequently, in this study, we carried out in-depth identification and examination of the Whirly gene family in *B*. *napus* (ZS11), utilizing both bioinformatics and comparative genomics approaches. We examined chromosomal placement, phylogenetic affiliations, and gene configurations, etc. Besides, the transformations that occurred within the Whirly gene family constituents throughout the development of *B*. *napus* were assessed. This study provides foundational insights into the functions and effects of the Whirly gene family in this plant species, facilitating future research into the molecular mechanisms that drive the evolution of genes in *B. napus* and its diploid ancestors. Our findings enhance understanding of the complex evolutionary dynamics and adaptations in these plants.

## 2. Methods

### 2.1. Genome-Wide Identification and Whirly Gene Family Analysis

To pinpoint the Whirly gene family members in *B*. *napus* and its diploid precursors (*B*. *rapa* and *B*. *oleracea*), we sourced the genome and protein reference sequences along with their respective annotation files for the three species from the BRAD database (http://www.brassicadb.cn, accessed on 6 March 2024) [33]. Additionally, *Whirly* sequences and related annotation data for *A*. *thaliana* were extracted from the TAIR database (https://www.arabidopsis.org, accessed on 7 March 2024) [34]. Our initial step involved securing the hidden Markov model (PF08536) that encodes the Whirly protein domain from the Pfam database (http://pfam-legacy.xfam.org, accessed on 9 March 2024) [35]. Using the hmmsearch tool (version 3.4) under the Linux environment, we searched for *B*. *napus* protein sequences in our database. Homology comparison was conducted through Blast (E value less than 1 × 10^−5^). After excluding sequences with low coverage, we utilized tools such as NCBI CD-search [36] and InterProScan (version 5.32-71.0) [37] to retrieve domains with Whirly protein features. Finally, candidate genes with Whirly domain features were retained for further analysis.

### 2.2. Protein Physicochemical Properties and Gene Structures Analysis

The physicochemical attributes of the discerned sequences were evaluated utilizing the Expasy website (https://web.expasy.org/protparam/, accessed on 17 March 2024) [38]. Subsequently, WoLF PSORT aided in forecasting the subcellular localization (https://wolfpsort.hgc.jp/, accessed on 18 March 2024) [39]. We explored the Whirly protein sequences’ conserved domains and motifs using the NCBI CD-search [36] and MEME Suite 5.5.0 online tools [40]. The architectural analysis of the genes was conducted employing the Gene Structure View feature in TBtools (version 1.120) [41].

### 2.3. Analysis of the Chromosomal Localization and Phylogeny of the Whirly Gene Family

Based on the genome information about *B*. *rapa*, *B*. *oleracea*, and *B*. *napus*, we plotted the chromosomal localization of *Whirly* genes using the online software MapInspect (version 1.0). The MEGA-X software (version 10.1.8) was used for the repeated alignment of Whirly protein sequences among ten species, i.e., *Z*. *mays*, *O*. *sativa*, *A*. *thaliana*, *M*. *sativa*, *Glycine max* L., *Brachypodium distachyon* L., *B*. *rapa*, *Sorghum bicolor* L., *B*. *oleracea*, and *B*. *napus* [42]. Then, a phylogenetic tree was constructed by the maximum likelihood (ML) method, with the Bootstrap value set at 1000, and iTOL (https://itol.embl.de/, accessed on 25 March 2024) was applied for visualization [43].

### 2.4. Protein 3D Structure Prediction and Cis-Acting Element Analysis

Using Phyre2 (http://www.sbg.bio.ic.ac.uk/phyre2/, accessed on 3 April 2024) [44], the three-dimensional protein structures were predicted through homology modeling by comparing the Whirly protein sequences. Then, these structures were visualized using Pymol (version 2.5.4) [45]. The sequence of 2 kb was extracted from the upper stream of the transcription start site in each gene through Perl Script, and then was uploaded to the PlantCARE (http://bioinformatics.psb.ugent.be/webtools/plantcare/html/, accessed on 8 April 2024) database for subsequent analysis on cis-acting elements [46].

### 2.5. Collinearity Analysis and Protein Interaction Network Prediction of Whirly Gene Family

MCscanX (version 1.1.11) was employed to investigate the gene duplication mechanisms and collinearity within the Whirly gene family in *B*. *napus* and its diploid progenitors [47]. To assess the rates of synonymous (Ks) and non-synonymous (Ka) substitutions among duplicated gene pairs, along with their ratio (Ka/Ks), we used the Simple Ka/Ks Calculator (NG) package available in TBtools (version 1.120) (Appendix A) [41]. Based on the protein–protein interaction data from the STRING database (https://cn.string-db.org/, accessed on 13 April 2024) [48], we constructed a network of Whirly protein interactions using the Whirly proteins from *A*. *thaliana* as a reference. The network analysis and the visualization of the data were conducted using Cytoscape software (version 3.9.1) [49].

### 2.6. Subcellular Localization Analysis

We first used PCAMBIA2300-GFP as a vector to clone and transform the candidate genes. The successful Agrobacterium cultures were diluted and used to infect *Nicotiana tabacum* L. leaves, marking the infected areas. The infected *N*. *tabacum* was then dark-cultivated overnight. After 2–3 days of infection, the infected areas were cut, the epidermis was peeled for slide preparation, and finally, the locations affected by the candidate genes were observed under a confocal microscope. GFP represents the green fluorescence field (488 nm), CHI represents chlorophyll autofluorescence, DAPI represents the DAPI field (358 nm), DIC represents bright field, and Merge represents the combined field.

### 2.7. Expression Patterns and qRT-PCR Analysis of Whirly Gene Family in B. napus

The Whirly gene family’s expression patterns in *B*. *napus* were visualized and analyzed using the Heatmap package in R (version 4.2.1) based on the data issued by Liu et al. [50] (http://yanglab.hzau.edu.cn/BnIR, accessed on 23 April 2024). These data were concerned with the amount of gene expression (TPM) in *B*. *napus* under different tissues (root, cotyledon, petal, sepal, filament, pollen, stem epidermis, vegetative rosette, and silique).

To further investigate the potential biological roles of the *Whirly* genes, we conducted qRT-PCR analysis on ZS11 seeds. These seeds were grown in a tissue culture chamber maintained at 25 ± 1 °C, with a 12:12 light/dark cycle. Once the seedlings reached the five-leaf stage, they were subjected to stress treatments, which included irrigation with 200 mmol/L sodium chloride and 10% PEG-6000, along with spraying the adaxial/abaxial leaf surfaces with 200 mmol/L methyl jasmonate. Leaf samples were collected at 0, 3, 6, 9, 12, and 24 h post-treatment, immediately frozen in liquid nitrogen, and stored at −80 °C. Each treatment included three biological replicates.

For RNA extraction, total RNA was isolated from the collected leaf tissues using a specific kit provided by Sangon Biotech Engineering (Shanghai, China). This RNA was then converted to cDNA using the PrimeScriptTM RT reagent kit from TaKaRa (Kusatsu, Japan). The expression levels of the *Whirly* genes were measured using the ABI-7500 fluorescence quantitative system along with Takara’s qPCR SYBR Green Master Mix. Specific primers were designed with the assistance of Primer 5 software. BnActin7, with GeneBank ID: GBEQ01027912.1, was used as the internal reference gene, as detailed in Appendix A. Each reaction was conducted in triplicate to ensure accuracy. The 2^−ΔΔCT^ method was used to analyze the expression levels of the *Whirly* genes across various treatments and time points, with the raw data available in Appendix A.

## 3. Results

### 3.1. Identification and Physicochemical Characterization of Whirly Genes

Taking the *Whirly* genes in *A*. *thaliana* as a model, the homologous alignment was performed in *B*. *rapa*, *B*. *oleracea*, and *B*. *napus* using various tools (Pfam, SMART, CD-search, etc.). Finally, three, four, and six *Whirly* genes were identified in these three species, respectively (Table 1). According to the physicochemical properties of these genes, the number of amino acids ranged between 95 (*BnAWHY3b*) and 269 (*BrWHY3a*); the molecular weight was measured to be 10643.99Da (*BnAWHY3b*)–29404.12Da (*BrWHY3a*); the isoelectric point was 4.91 (*BnAWHY3b*)–10.22 (*BoWHY4*). Most Whirly family members were basic amino acids. In addition, all proteins were unstable, except for *BrWHY1a*, *BoWHY4*, and *BnAWHY1b*, which were defined as stable proteins due to their instability indexes of less than 40 (39.52, 39.28, and 27.58, respectively).

### 3.2. Chromosomal Localization of the Whirly Genes

Using MapInspect, chromosomes were localized based on the genome annotation files concerning *B*. *rapa*, *B*. *oleracea*, and *B. napus*. We also found that 13 *Whirly* genes were localized on 12 chromosomes (Figure 1). Three *BrWHY* genes were localized on three chromosomes (Ar06, Ar07, and Ar08) in the genome of *B*. *rapa*, and four *BoWHY* genes were localized on three chromosomes (Co06, Co07, and Co08) in the genome of *B*. *oleracea*. Comparing the distribution patterns of *Whirly* genes in A and C subgenomes of *B*. *napus* and in genomes of *B*. *rapa* and *B*. *oleracea*, we found that these genes were homologous, with the same relative locations. Moreover, one gene was lost during the formation of *B*. *napus* due to the incomplete assembly of the Cn07 chromosome, while the other six (*BrWHY1a* and *BnAWHY1b*, *BrWHY2a* and *BnAWHY2b*, *BrWHY3a* and *BnAWHY3b*, *BoWHY1a* and *BnCWHY1b*, *BoWHY2a* and *BnCWHY2b*, *BoWHY3a* and *BnCWHY3b*) maintained their relative locations on chromosomes in this process. Based on these results, we deduced that there was no significant change in the *Whirly* genes of *B*. *napus* that originated from its diploid ancestors during hybridization and polyploidization.

### 3.3. Phylogenetic and Homology Modeling Analysis of Whirly Proteins

To further analyze the phylogenetic relationship in the Whirly gene family, a phylogenetic tree was constructed by referring to 34 Whirly protein sequences in 10 types of monocots and dicots (Figure 2). The 34 proteins were classified into six subgroups (I~VI). Subgroup IV has the largest number of members, which are *BnAWHY1b*, *BnAWHY3b*, *BnCWHY2b*, and *BnCWHY3b*; subgroup I contains *BnAWHY2b* and *BnCWHY1b*; subgroup III shows a close genetic relationship of members between soybean and alfalfa. In this tree, the Whirly proteins in monocotyledon clustered in subgroups II, V, and VI, while those in dicotyledon clustered in others. The Whirly proteins in *B*. *rapa* and *B*. *oleracea* gathered with the homologous ones in *B*. *napus* in subgroups I and IV, showing a good phylogenetic relationship. These results implied the highly conserved Whirly proteins in monocotyledon and dicotyledon.

Based on the homology modeling and prediction of Whirly protein sequences from *A*. *thaliana*, *B*. *rapa*, *B*. *oleracea*, and *B*. *napus*, we found that the Whirly proteins in *B*. *napus* and its diploid ancestors, except for BoWHY4, BnAWHY1b, and BnAWHY3b, have a highly similar three-dimensional structure to that of AtWHY proteins (Figure 3) [51]. The Whirly family members are composed of three α-helices (α1–α3) and eight β-sheets (β1–β8), of which groups β1–β4 and β5–β8 are distributed perpendicularly and connected by the α1-helix. In this family, the three-dimensional structures of BoWHY4, BnAWHY1b, and BnAWHY3b are not exactly the same as other members. These similarities and differences can explain the reasons for similar or different functions among Whirly family members.

### 3.4. Motif Composition, Conserved Structural Domains, and Gene Structures of the Whirly Gene Family

The *Whirly* genes were compared in terms of gene structures, motifs, and conserved domains to reveal their phylogenetic relationship in diploid *B*. *rapa*, *B*. *oleracea*, and in allotetraploid *B*. *napus* (Figure 4). Figure 4a illustrates that members of the Whirly family contain between three and nine motifs. Most homologous *Whirly* sequences have similar, but not identical, motif compositions, probably because of the functional diversity and evolutionary mutation of *Whirly* genes. Furthermore, although the Whirly family members are composed of different conserved motifs in *B*. *napus* and its diploid ancestors, there are three overlapped ones (Motif 2, Motif 3, and Motif 5), indicating that these three motifs are probably of great importance (Figure 4a,d). The analysis on conserved domains shows that the pfam08536 domain, which is symbolic in the Whirly gene family, is contained in *BrWHYs*, *BoWHYs*, and *BnWHYs* (Figure 4b). By comparing conservative motifs and domains, we found that most of the conserved domains in the Whirly gene family contain Motif 1, Motif 4, and Motif 2, suggesting that these three motifs may be unique to the Whirly gene family. Their composition and arrangement order are of significant importance for their functional exercise (Figure 4a,b). Additionally, Motif 3, Motif 5, Motif 6, Motif 7, Motif 8, Motif 9, and Motif 10 are not present in the conserved regions of the Whirly gene family.

A gene family evolves mainly because of its diverse gene structures [52,53]. In this study, we analyzed the intron–exon structure to understand the structural characteristics of identified *Whirly* genes. According to Figure 4c, most homologous *Whirly* genes have similar structures in *B*. *rapa*, *B*. *oleracea*, and *B*. *napus*, while several others are obviously different. This is explicated as follows: compared to *BrWHY3a*, *BnAWHY3b* has four exons and four introns reduced in *B*. *napus*; compared to *BrWHY1a*, *BnAWHY1b* has two exons and two introns reduced; and compared to *BoWHY1a*, *BnCWHY1b* has one exon and one intron reduced. Besides, despite similar gene structures between *BrWHY2a* and *BnAWHY2b*, the latter’s third exon is reduced by 3bp. These findings indicate that some introns/exons may be lost during the evolution of the Whirly gene family in *B*. *napus*, which can be used to explain the functional difference among different family members. Moreover, in this species, the A subgenome may be more susceptible than the C. Based on the conserved domain characteristics of the Whirly gene family and genome annotation files, we found that in the gene structure, the third to sixth exons of most members of the family are relatively conserved. However, only a few members show a lack of conserved regions (*BoWHY4*, *BnAWHY1b*, *BnAWHY3b*), which may be due to the diversity of *Whirly* gene functions and mutations that occurred during evolution, providing clues for existing functional differences. These conserved exon structures may be more meaningful for their potential biological functions. Furthermore, we also found that compared to subgroup I, members of subgroup IV exhibit instability in conserved regions, which also suggests that different members of the Whirly gene family may have different functions.

### 3.5. Cis-Acting Elements in the Promoter Region of the Whirly Gene Family

Considering the potential impact of diverse cis-acting elements within the gene promoter on gene functionality, we examined the sequences located 2 kb upstream of the prospective genes. Our analysis led to the identification of 11 distinct categories of cis-acting elements connected with various aspects of plant physiology, as illustrated in Appendix A. These elements are implicated in processes such as plant growth, development, photoreactivity, stress response, and hormonal activities. Within this context, *BrWHY3a* and *BnAWHY3b* are distinguished by possessing the highest count of cis-acting elements, totaling 31, while *BoWHY4* is noted for having the least (13). The element (ARE) associated with anaerobic induction was detected in all genes; the two elements (ABRE and CAT-box) participating in the abscisic acid signal transduction and meristem-related gene expression were mostly involved in diploid *B*. *rapa* and *B*. *oleracea* but were less involved in the *Whirly* gene promoters of tetraploid *B*. *napus*. It should be noted that the cis-acting elements identified in most *Whirly* gene promoters are predominantly related to photoresponse (Box4, GT1-motif, and G-box), anaerobic induction (ARE), and methyl jasmonate (CGTCA-motif and TGACG-motif).

### 3.6. Collinearity Relationship of the Whirly Gene Family

It has been verified that gene duplication is crucial to the expansion of gene families and the evolution of plant genomes [54]. We analyzed the collinearity of *Whirly* genes to explore the duplication events in which *B*. *napus* and its ancestral species participated (Figure 5). Based on the comparison of collinearity relationship in each species, we detected one pair of collinear genes in each diploid ancestor and seven pairs in *B*. *napus*. Then, this relationship was compared among these three species. It was found that *B*. *napus* had 10 pairs of collinear genes with *B*. *rapa* and *B*. *oleracea*, respectively, with these genes showing characteristics of segmental duplication. This implied that segmental duplication conduced to the evolution of the Whirly gene family. To better understand whether selection pressure was correlated with *Whirly* genes after duplication events, we calculated the synonymous (Ks) and non-synonymous (Ka) substitution rates of duplicate gene pairs and the ratio (Ka/Ks) between them (Appendix A). The calculation results showed that the value of Ka/Ks was less than 1 in all pairs, ranging between 0.127 and 0.710, which proved that the *Whirly* genes in *B*. *napus* and its diploid ancestors underwent strong purifying selection after duplication. Besides, the duplicate gene pairs were duplicated 3.33~28.50 million years ago, indicating that Whirly may be an ancient gene family and was highly conserved during its evolution.

### 3.7. Prediction of Whirly Protein Interactions Network in B. napus

Using the homologous proteins in *A. thaliana* as a model, we constructed an interaction network for Whirly proteins in *B*. *napus* in the online database STRING, which was conducive to better understanding the biological functions of these proteins and the dynamic control of various molecules (Figure 6). In this network, the edge number was expected to be 12; the enriched *p*-value was set at 6.66 × 10^−6^. As shown in Figure 6, two Whirly family members in *B*. *napus* were involved in potential interactions. This network contains six proteins maintaining the plastid stability, four DNA repair proteins, four proteins concerning the chloroplast functions, two fructokinases, one galactitol synthetase, one ubiquitin carboxy-terminal hydrolase, one serine carboxypeptidase, and one DNA methyltransferase, which constitutes a total of 104 interactions. BnAWHY1b and BnCWHY3b are essential to the whole network. Furthermore, this network also includes eight key nodal proteins. Therefore, we predicted the direct or indirect interactions among different Whirly family members in *B*. *napus* and other important interacting proteins in this network.

### 3.8. Subcellular Localization of BnAWHY1b and BnCWHY3b

To explore the potential biological functions of BnAWHY1b and BnCWHY3b, we conducted subcellular localization analysis. As shown in Figure 7, the BnAWHY1b protein displayed significant fluorescence signals in both the cell membrane and nucleus, but not in other areas. In contrast, the BnCWHY3b protein showed fluorescence in the chloroplasts. These results suggest that different members of the same family may function in different cellular components. Specifically, BnCWHY3b may play a key role in chloroplasts.

### 3.9. Tissue Expression Patterns of the Whirly Gene Family in B. napus

To further reveal the expression patterns and potential biological functions of identified *BnWHY* genes, we analyzed the expression of *Whirly* genes in *B*. *napus* under different tissues based on the RNA-seq data (Figure 8). The results showed that most *Whirly* genes were highly expressed (TPM > 10) in roots, stem epidermises, and rosette leaves. *BnAWHY2b*, *BnCWHY1b*, and *BnCWHY3b* presented similar expression patterns both in these tissues (Figure 8a) and in the development of silique (Figure 8b). Notably, these genes showed higher expressions (TPM > 20) in the initial stage of silique development (2DAF-10DAF). Furthermore, as silique kept growing and developing, *BnCWHY3b* still maintained a higher expression level (TPM > 10), which implied that this gene made great contributions to the growth and development of *B*. *napus*.

### 3.10. Expression Pattern Analysis of BnWHY Genes by qRT-PCR

Due to the presence of abundant hormone-responsive (CGTCA-motif, TGACG-motif, ABRE) and abiotic stress-responsive (TC-rich repeats, ARE) cis-elements in the *Whirly* gene promoter, we analyzed the expression patterns of this family under various hormones and abiotic stresses to further explore the potential functions of the Whirly gene family (Figure 9). The results indicate that under salt stress, all members of the Whirly gene family exhibit an initial upregulation followed by downregulation (Figure 9a). Specifically, compared to the control (0 h), members of subfamily I (*BnAWHY2b*, *BnCWHY1b*) reach a higher expression level 12 h after stress treatment, especially *BnCWHY1b*, which shows the most significant upregulation. Members of subfamily IV (*BnAWHY1b*, *BnAWHY3b*, *BnCWHY2b*, *BnCWHY3b*) similarly reach peak expression levels 6 h post-treatment. Evidently, the *Whirly* genes in *B*. *napus* exhibit a positive response to salt stress. However, their expression tends to decrease with prolonged treatment. In comparison to salt stress, under PEG stress, members of the Whirly gene family also show an up-then-down expression trend (Figure 9b). Notably, members from subfamilies I and IV reach their peak expression 9 h post-treatment, with *BnAWHY1b* being the most upregulated. This suggests that *BnAWHY1b* from the Whirly gene family might play a vital role in *B*. *napus*’ response to PEG stress. Under MeJA stress, members of subfamily I (*BnAWHY2b*, *BnCWHY1b*) differ markedly in their expression pattern compared to other genes, primarily showing an initial upregulation followed by downregulation, peaking 6 h post-treatment (Figure 9c). However, most members of subfamily IV (*BnAWHY1b*, *BnCWHY2b*, and *BnCWHY3b*) mainly exhibit a downregulation trend. Contrasting with *BnAWHY1b*, *BnCWHY2b*, and *BnCWHY3b* of subfamily IV, *BnAWHY3b* shows an up-then-down trend, peaking 9 h post-treatment. Moreover, the results demonstrate that members of the Whirly gene family in *B*. *napus* (subfamilies I and IV) have similar response patterns under both salt and PEG stress (Figure 9a,b), primarily characterized by an up-then-down trend. However, under hormone stress, they exhibit different expression patterns (Figure 9c). This variation might be associated with their involvement in distinct biological processes or the execution of different biological functions, hence the specificity.

## 4. Discussion

The process of polyploidization, a recurring event in species evolution, plays a crucial role in plant diversification and the emergence of new species. It not only influences evolutionary trajectories but also enhances the adaptability of species to diverse environmental conditions [55,56,57]. Plants that have experienced polyploidization will undergo changes in genome size (such as genome rearrangement and homologous gene loss) and gene sequence [3]. The successful sequencing and assembly of genomes in *B*. *napus* and its diploid ancestors facilitate the genome-wide identification of *Whirly* genes and the exploration of their potential biological functions [58,59,60]. The Whirly gene family plays a major part in regulating leaf senescence and silique development, stabilizing plastids and mitochondria, and repairing genomic damages. It has been reported in *A. thaliana*, *O. sativa*, *Z. mays*, *S. tuberosum*, and *M. sativa*, but is seldom studied in *Brassica*. Therefore, analyzing the Whirly gene family within the *B*. *napus* genome and contrasting it with the homologous genes of its diploid predecessors provides deeper insights into the evolutionary trajectory during the emergence of the tetraploid *B*. *napus*.

In our research, homologous alignment and conserved domain methods led to the discovery of three, four, and six *Whirly* genes within *B*. *rapa*, *B*. *oleracea*, and *B*. *napus*, respectively. Notably, a gene was absent in *B*. *napus* due to an incomplete assembly of the Cn07 chromosome. The remaining genes in the A and C subgenomes align closely in number and chromosomal distribution with the *Whirly* genes in the genomes of its diploid predecessors, suggesting minimal gene loss in the *B*. *napus* Whirly gene family. A study of subcellular localization revealed a predominance of these genes in the chloroplasts of the three species, underscoring their significant role, particularly in safeguarding and repairing chloroplast genomes. This finding is consistent with earlier identifications of Whirly proteins, affirming their localization in chloroplasts and involvement in the reparative processes of chloroplast genomes [11]. The phylogenetic relationship among different species indicated that the Whirly gene family in monocotyledon and dicotyledon separately clustered in different branches. This finding aligns with earlier research, suggesting a high level of conservation of this family throughout evolution [25]. A more detailed examination of the conserved motifs and gene structures of this family reveals the presence of Motif 2, Motif 3, and Motif 5 in most *Whirly* genes. Notably, distinctions in *BnAWHY1b*, *BnAWHY3b*, and *BnCWHY1b* between *B*. *napus* and its diploid forebears are observed, marked by the absence of several conserved motifs. Meanwhile, some introns/exons were also lost in gene structures, which was caused by the functional diversity and evolutionary mutation of *Whirly* genes. In addition, this diversity was promoted by introns/exons due to their varying structures, which helped understand the functional difference among different genes.

The genes, containing various types of cis-acting elements, are different in biological functions [61]. In this investigation, we discerned specific elements within the promoters of *Whirly* genes in *B*. *rapa*, *B*. *oleracea*, and *B*. *napus*. Notably, *B*. *napus* had fewer cis-acting elements (151) compared to *B*. *rapa* and *B*. *oleracea* (176). However, the count of photoresponse-related elements was comparable between *B*. *napus* (57) and its diploid precursors (58). This discrepancy primarily originated from variations in the cis-acting elements associated with plant growth, development, hormone responses, and stress responses, particularly those linked to abscisic acid signal transduction (ABRE) and meristem (CAT-box). Our findings supported the notion that gene duplication significantly contributes to the proliferation of gene families and the evolution of plant genomes [54]. Our data depicted a segmental duplication pattern characterizing the collinearity between *B*. *napus* and its diploid forebears. A Ka/Ks ratio below 1 highlighted a robust purifying selection imposed on the *Whirly* genes in these species post-duplication, indicating potential correlations between selection pressure and these genes following duplication. Moreover, since the Ks value in duplicated genes often acts as a molecular clock, uniformity in its values amongst these genes is anticipated over time. Furthermore, we determined that the *Whirly* genes in *B*. *napus* and its diploid ancestors underwent duplication approximately 3.33 to 28.50 million years ago. This evidence suggests the ancient origins of the Whirly gene family and underscores its significant conservation throughout evolutionary history.

The change in gene expression patterns can directly reflect gene functions. In our research, most of the *Whirly* genes in *B*. *napus* exhibited expression in roots, stem epidermises, and rosette leaves (TPM >10). However, the organizational structure and expression patterns were fundamentally consistent within the same subgroup. *BnAWHY2b*, *BnCWHY1b*, and *BnCWHY3b* presented similar expression patterns (Figure 8a), which was also the truth during the development of silique (Figure 8b). Moreover, with the continuous growth and development of silique, *BnCWHY3b* was still highly expressed (TPM > 10), which implied that this gene may greatly contribute to the growth and development of *B*. *napus*. Previous research has reported that plants can reach the stress responses by increasing the transcriptional level of *Whirly* genes, and they can become more tolerant to stresses through the interactions between Whirly family members. By analyzing the expression patterns of the *Whirly* genes in different tissues, it was found that *BnCWHY3b* has a higher expression level in different tissues, especially during the growth and development of silique (Figure 8). This also implies that *BnCWHY3b* may play an essential role during silique maturation. Further analysis of the *Whirly* genes’ expression patterns under various abiotic stresses through qRT-PCR revealed that subfamilies I and IV respond similarly under salt and PEG stress (Figure 9), mainly showing an initial upregulation followed by downregulation. However, they display different expression patterns under hormone stress, which might be related to their involvement in different biological processes or performing various biological functions. Additionally, we found that the *Whirly* genes actively respond to drought stress, especially *BnAWHY1b*, suggesting that *BnAWHY1b* might play a significant role in *B*. *napus* coping with drought stress. This is consistent with previous research results where exogenously applied hydrogen peroxide aids in the accumulation of *WHY1* in chloroplasts [54]. Interestingly, in the analysis of cis-acting elements, we identified numerous antioxidant-related elements (AREs). However, how they function, whether they provide protection to the plant, and the specific mechanisms of action still need further investigation in the future.

## 5. Conclusions

This study employed bioinformatics and comparative genomics approaches to analyze the *Whirly* genes in allotetraploid *B. napus* and its diploid progenitors. We identified six *Whirly* genes in *B. napus*, three in *B. rapa*, and four in *B. oleracea*. Whirly proteins are categorized into six subgroups based on sequence characteristics and exhibit high conservation across both monocots and dicots. Predictive analyses of cis-acting elements in the promoter regions of the Whirly gene family indicate that many of these elements are linked to light response, anaerobic induction, and methyl jasmonate responsiveness. Elements associated with abscisic acid signal transduction and meristem activity were predominantly found in the diploids *B. rapa* and *B. oleracea* but were less prevalent in the tetraploid *B. napus*. Gene structure analysis indicated a potential loss of introns/exons in the evolutionary history of the Whirly gene family in *B. napus*, likely influenced by segmental duplication as suggested by the collinearity relationships. The *Whirly* genes in *B. napus* and its ancestors appear to have undergone strong purifying selection post-duplication. Additionally, Motif 1, Motif 4, and Motif 2 exhibit high conservation within this gene family. Through protein–protein interaction networks and subcellular localization analysis, BnAWHY1b and BnCWHY3b are identified as potential core members of the Whirly gene family. Further studies on their protein–protein interactions and co-expression relationships could help elucidate their functions. Expression analysis showed that patterns within the same subfamily were consistent, supported by tissue-specific expression data. Further qRT-PCR analysis indicated that the *BnAWHY1b* gene could be particularly important in *B. napus* for responding to abiotic stress, especially under drought conditions. This study, through bioinformatics and comparative genomics analysis of the Whirly gene family in *B. napus* and its diploid ancestors, provides insights into the potential biological functions of this gene family and the occurrence of plant polyploidization events.

## Figures and Tables

**Figure 1 plants-13-02243-f001:**
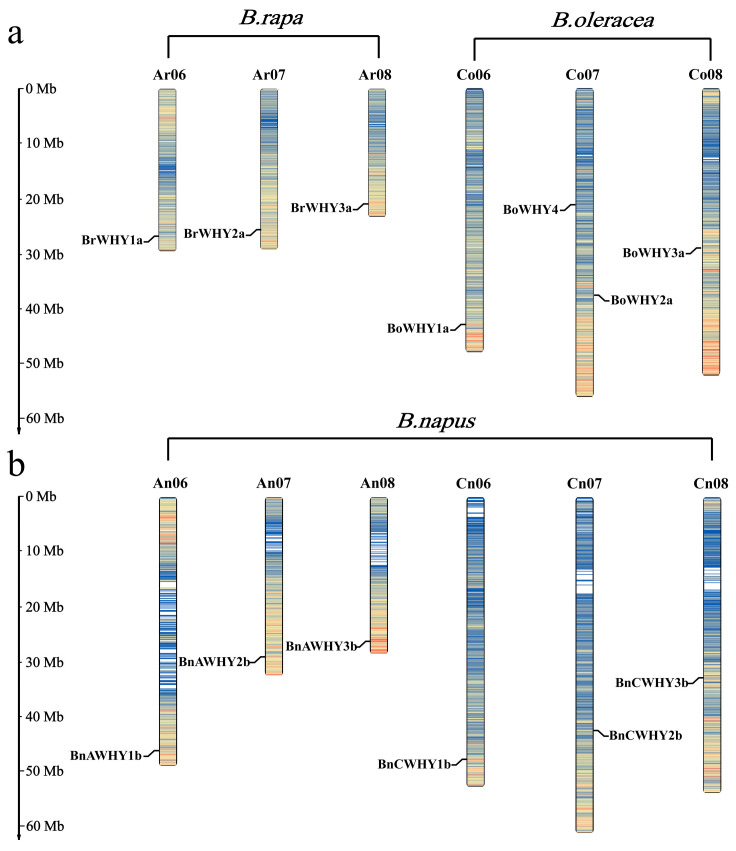
Chromosomal location of the Whirly gene family in *B*. *rapa*, *B*. *oleracea* (**a**), and *B*. *napus* (**b**).

**Figure 2 plants-13-02243-f002:**
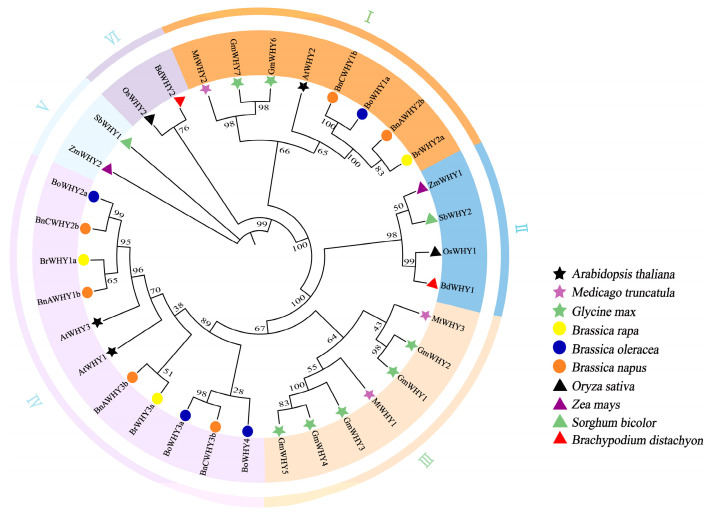
Evolutionary analysis of the Whirly gene family. Utilizing the protein sequences from 10 variations of monocots and dicots, an evolutionary tree for the Whirly gene family was established employing the maximum likelihood (ML) technique, substantiated by 1000 bootstrap replicates.

**Figure 3 plants-13-02243-f003:**
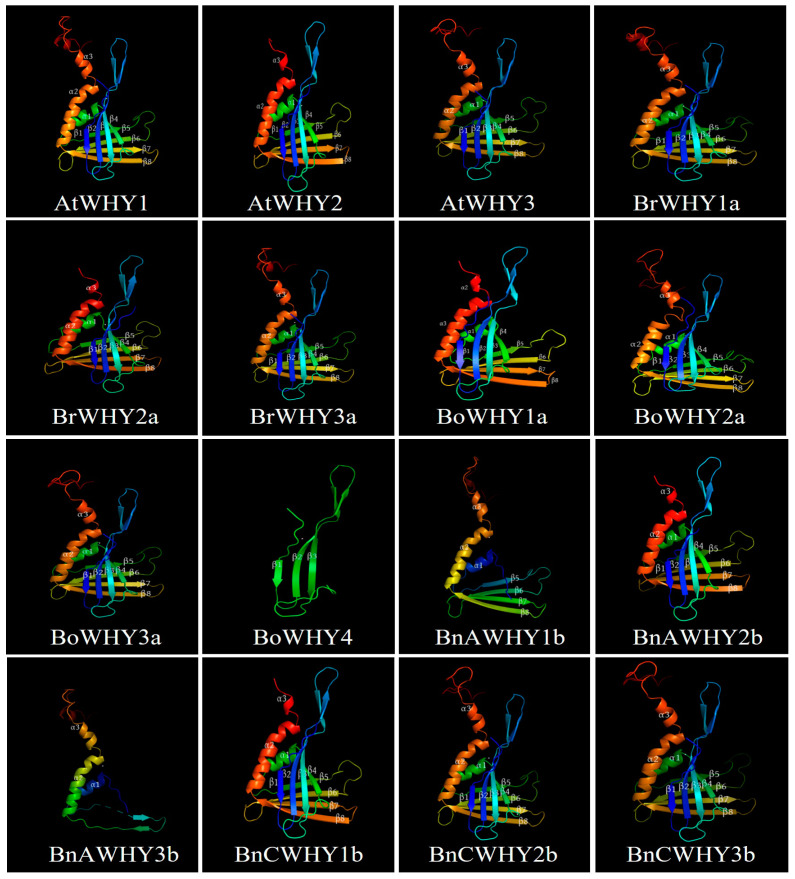
The predicted three-dimensional structure of Whirly proteins in *A*. *thaliana* (AtWHY1–3), *B*. *rapa* (BrWHY1a–3a), *B*. *oleracea* (BoWHY1a–3b), and *B*. *napus* (BnAWHY1b–3b, BnCWHY1b–3b), which primarily includes α-helices (α1–α3) and β-sheets (β1–β8), with the number of structures indicated numerically.

**Figure 4 plants-13-02243-f004:**
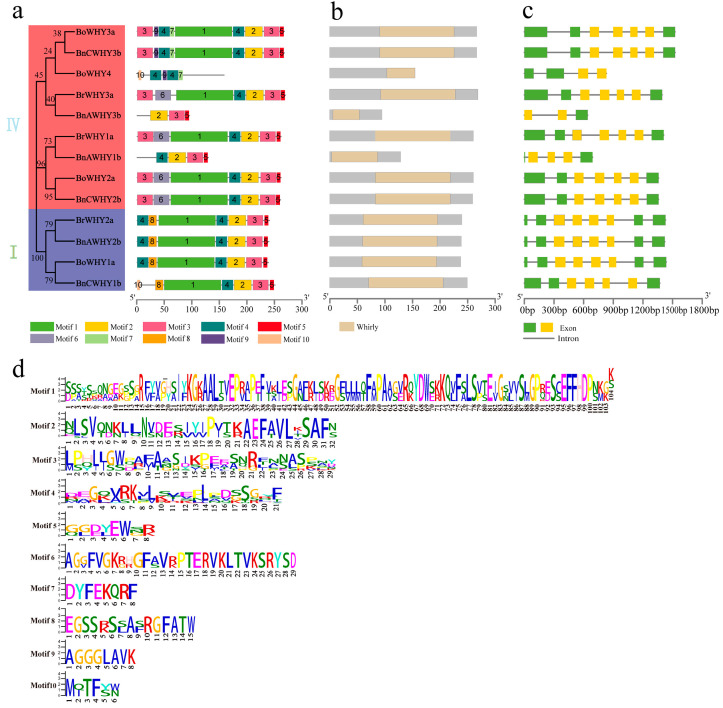
Gene structure and conserved motif analysis of Whirly gene family. (**a**) The 10 conserved motifs have different colors and sequences, and the different motifs correspond to one another in the structure. (**b**) Domains of the Whirly gene family. (**c**) The exon–intron structure of family members. (**d**) Logo of each motif.

**Figure 5 plants-13-02243-f005:**
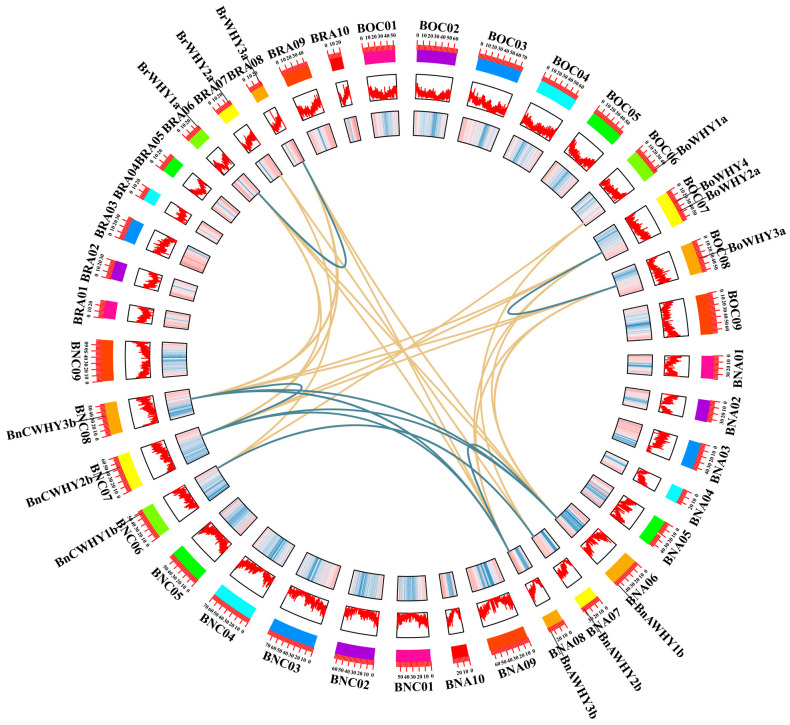
Collinearity analysis of the Whirly gene family. BRA, BOC, and BN represent the chromosomes of *B*. *rapa*, *B*. *oleracea*, and *B*. *napus*, respectively. Members of the Whirly family with gene replication events were all located on corresponding chromosomes. The yellow lines connect collinearity between species, and the blue lines connect collinearity within species.

**Figure 6 plants-13-02243-f006:**
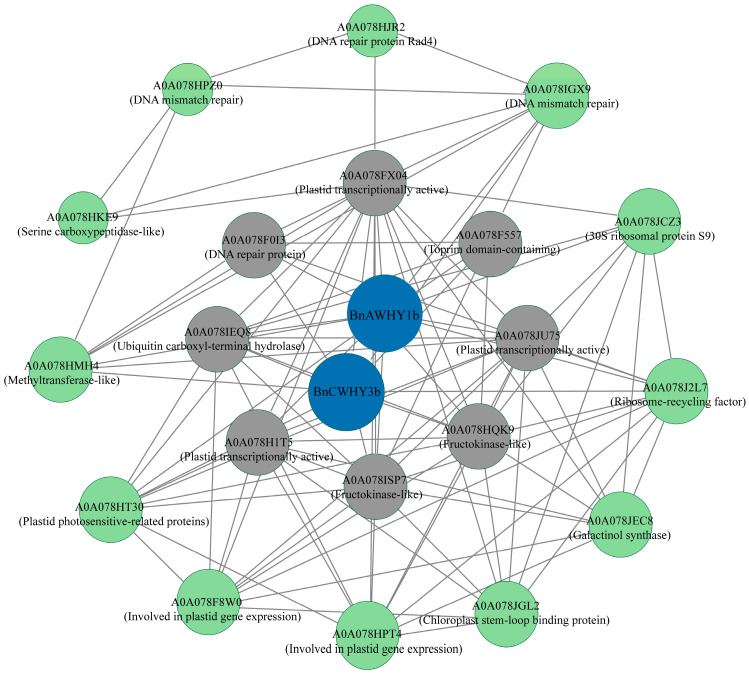
Mapping interactions among Whirly proteins in *B*. *napus*. Nodes represent different proteins, and lines represent interactions between proteins. In Cytoscape, they are arranged according to the degree algorithm (degree = 1–18), where blue circles represent members of that gene family. Gray circles represent proteins that have strong interactions with it, and green circles represent proteins that may interact with it, albeit with weaker relationships.

**Figure 7 plants-13-02243-f007:**
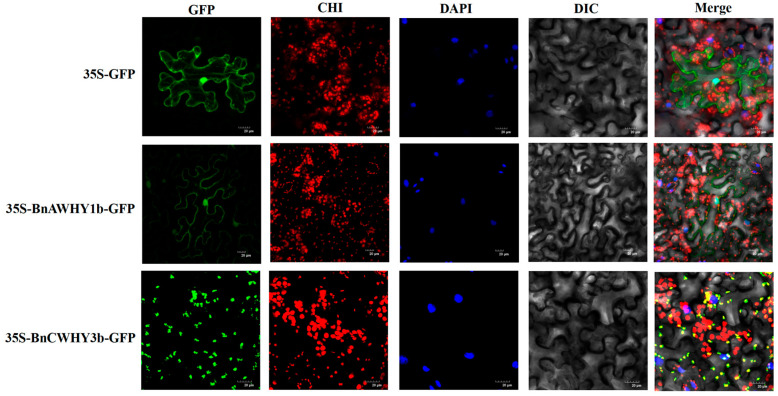
Subcellular localization analysis of BnAWHY1b and BnCWHY3b proteins in *N. tabacum* leaves. The proteins 35S-GFP, 35S-BnAWHY1b-GFP, and 35S-BnCWHY3b-GFP were transiently transformed into *N*. *tabacum*. GFP represents the green fluorescence field, CHI represents the autofluorescence field of chloroplasts, DAPI represents the DAPI field (a type of nuclear stain), DIC represents the bright field, and Merge represents the composite field. GFP field: 488 nm, DAPI field: 358 nm, CHI field: 488 nm. Note that the excitation light for green fluorescence and chloroplast autofluorescence is the same, but the collected light wavelengths are different. For reference, the scale bar represents a length of 20 μM, facilitating size assessment.

**Figure 8 plants-13-02243-f008:**
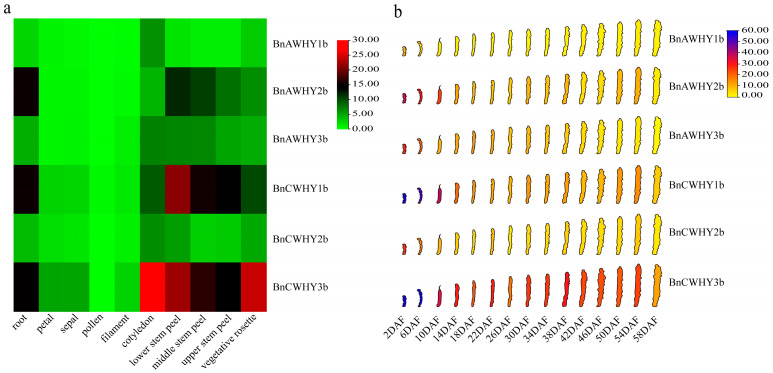
Tissue expression patterns of the Whirly gene family in *B*. *napus*. (**a**) Tissue expression pattern of the Whirly gene family in *B*. *napus*. (**b**) Expression patterns of the Whirly gene family in silique at different developmental stages, where DAF refers to days after flowering, and 2DAF refers to 2 days after flowering.

**Figure 9 plants-13-02243-f009:**
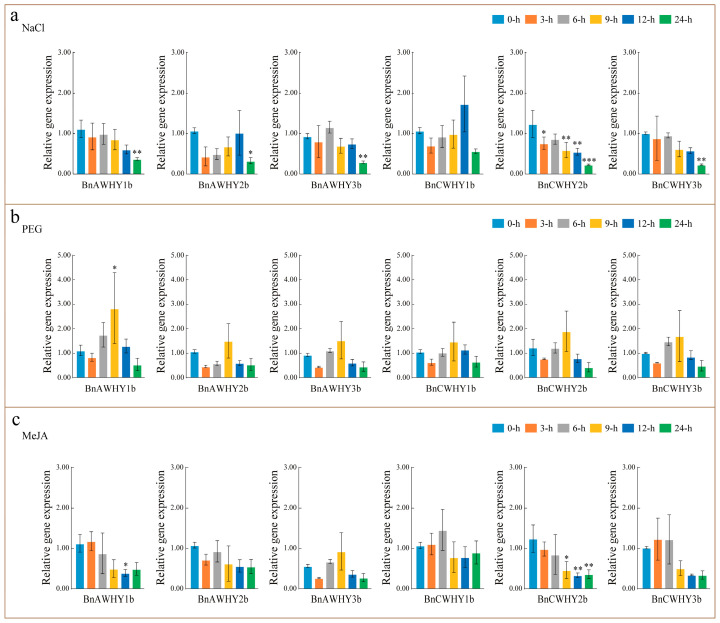
Expression patterns of the Whirly gene family in *B*. *napus* leaves under (**a**) the NaCl treatment; (**b**) the drought treatment; (**c**) the MeJA treatment. The expression levels of *Whirly* genes are relative to actin. The relative expression levels of each gene were calculated by the 2^−∆∆Ct^ method. The mean (±SE) expression values were calculated from three independent biological replicates and three technical replicates (*, *p* < 0.05; **, *p* < 0.01; ***, *p* < 0.001).

**Table 1 plants-13-02243-t001:** Physicochemical properties and basic information of *Whirly* genes in *B*. *napus* and its diploid ancestors.

Gene Name	Gene ID	Chromosome	Number of Amino Acid (aa)	Molecular Weight (Da)	Isoelectric Point (pI)	Instability Index	Orthologous Gene
*BrWHY1a*	*BraA06g039670.3C*	A06	261	28,804.77	9.56	39.52	*AT2G02740*
*BrWHY2a*	*BraA07g035880.3C*	A07	240	26,097.76	9.46	47.03	*AT1G71260*
*BrWHY3a*	*BraA08g030230.3C*	A08	269	29,404.12	9.30	47.37	*AT1G14410*
*BoWHY1a*	*BolC06g042680.2J*	C06	238	26,120.87	9.48	43.55	*AT1G71260*
*BoWHY2a*	*BolC07g032390.2J*	C07	261	28,763.65	9.58	44.89	*AT2G02740*
*BoWHY3a*	*BolC08g025140.2J*	C08	267	29,185.91	9.30	45.73	*AT1G14410*
*BoWHY4*	*BolC07g016070.2J*	C07	155	17,246.77	10.22	39.28	*AT1G14410*
*BnAWHY1b*	*BnaA06G0401800ZS*	chrA06	129	14,290.28	6.10	27.58	*AT2G02740*
*BnAWHY2b*	*BnaA07G0324700ZS*	chrA07	239	26,033.72	9.46	48.24	*AT1G71260*
*BnAWHY3b*	*BnaA08G0272100ZS*	chrA08	95	10,643.99	4.91	40.17	*AT1G14410*
*BnCWHY1b*	*BnaC06G0379900ZS*	chrC06	250	27,481.34	9.18	41.95	*AT1G71260*
*BnCWHY2b*	*BnaC07G0279600ZS*	chrC07	260	28,635.52	9.58	45.02	*AT2G02740*
*BnCWHY3b*	*BnaC08G0228600ZS*	chrC08	267	29,225.93	9.30	44.37	*AT1G14410*

## Data Availability

All data created or analyzed during this study are included in this article and its Appendix A. The sequences discussed are available on Ensembl Plants (http://plants.ensembl.org/index.html, accessed on 6 March 2024).

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
