# Peer review of "Genome-Wide Identification and Molecular Evolutionary History of the Whirly Family Genes in Brassica napus"

_plants, 2024, doi:10.3390/plants13162243_

Round 1

Reviewer 1 Report

Comments and Suggestions for Authors

General statements:

The study done here to understand the function, effects, and evolutionary history of the Whirly gene family in Brassica napus was detailed and informative. The methods were appropriately chosen, the analysis was thorough, and the paper was well-structured. Clearly a lot of good work has been put into this paper. I am satisfied with the content of the paper overall, but several corrections need to be made before this paper can be published.

1)      On line 100 (Section 2.1), Brassica rapa is misprinted as B. rara. Please go through the document to make sure all scientific names are reported correctly.

2)      The italicisation of et al. is not consistent in the paper. Some instances are italicised and others are not. Please make this consistent.

3)      There are several careless mistakes in the paper that need to be fixed such double spaces. Additionally, some full stops have mistakenly been used instead of commas, such as on line 277.

4)      Most of the programs in the Methods section do not have version numbers and online databases don’t have access dates. I suggest adding these.

5)      The Methods section is filled with superfluous text (e.g. lines 107 onwards to the end of section 2.1). The information there is fine, but I suggest rewriting these paragraphs to be more concise.

6)      On line 295, It is stated that the ten pairs of colinear genes show characteristics of segmental duplication but it’s not evident in the text or the figure how the authors came to this conclusion. Could you please clarify this point?

7)      I suggest that you add to the description of Figure 7 what the colours mean: e.g. grey indicates nodal proteins. Currently it is not clear what the colours mean.

8)      It looks to me that the start of section 3.8 (lines 327-333) is just reiterating what was written in the Methods section without adding new information. I suggest removing this and just including the results.

9)      It is not clear which development stages are being reported in Figure 9b. Could you please add a little more to the description about the development scale (i.e. what does 2DAF stand for).

Comments on the Quality of English Language

1)      The authors use extensive writing transitions (e.g. therein, thereinto, therefore, furthermore) but do not always use these transitions correctly. Could you please check the meaning of these transitions and make sure you are using them appropriately? For example, in the introduction, the authors use ‘further’ instead of ‘furthermore’ and on line 294.

2)      The abstract contains several unclear sentences (such as line 25 and 27) that could be removed or rephrased to flow better.

3)      The first reported result in the abstract: 'we identified six Whirly genes, with three and four found in its diploid ancestors' isn’t clear and could be rephrased.

4)      There are some unusual grammar choices in the Results such as ‘with a finding’ on line 196. Additionally, some of the sentences are not clear and could be rephrased (e.g. line 232 and line 254).

5)      Commas on line 256 are used excessively. I suggest splitting the points into different sentences to better convey the results.

Overall, the paper is understandable but wording choices has made sections of this paper confusing. The grammar used throughout this paper needs some moderate editing to get up to publication-standard. 

Author Response

Thank you very much for taking the time to review this manuscript. Please find the detailed responses below and the corresponding revisions shaded in a yellow-highlighted background in the re-submitted files.

Reviewer 2 Report

Comments and Suggestions for Authors

The reviewed manuscript, titled “Genome-Wide Identification and Molecular Evolutionary History of the Whirly Family Genes in Brassica napus” by Wang et al. is a work that is providing insight into the Whirly motif containing gene families in this economically relevant crop.  Overall, the work is interesting, and the authors used a series of previously published datasets and tools in order to provide some predictions about this gene family.  They then follow up this work with some very nice gene expression and protein localization studies for several of these members.  The wet lab work is important and very well done, and I believe this would be of interest to a variety of researchers.  I do think that the manuscript as written needs quite a bit more work prior to being published in its final form.  I look forward to seeing the authors do this.

Comments:

Line 237: the figure legend for figure 3 is noticeably briefer and less informative than other figure legends.  Additionally, the figure legend states that the figure contains the 3D structures of the Whirly proteins – when it contains the predicted 3D structures, at least as far as this reviewer can determine.  If any of these were empirically determined (e.g. crystallography, cryo-EM, etc) that should be made clear.  As written, it reads like the solid-state structure as depicted has not been published/solved.  I would insist that the authors make this distinction and clarify the figure and verbiage in this section accordingly.

Line 264: this is a comment that is more relevant for the preparation of the final, published work.  Figure four has a significant amount of information.  In the review copy of the manuscript the formatting is off, which makes it quite challenging to read and properly interpret.  In part A the authors present a motif analysis and in B they provide a domain analysis.  Perhaps it would make sense to merge this, providing insight to what motifs are present and absent in the various domains of the gene family?  Especially as the only motif identified is the one that is characteristic of this protein structural family.  Lastly, figure 4C provided the genic structure of the CDS for each.  The overall conclusions that the authors provide are simply that there are some introns and exons that differ as a part of the evolution of this family during specialization.  This does not seem to provide relevant insight – perhaps the exons should be color coded to show what is conserved and differs between each RNA species?  What differences are present due to alternative splicing in each of these genes?  Are they all simply constitutively spliced?  Is there any reported variants and/or edited isoforms that are reported?  E.g. A-I, perhaps C-U?  This part of the work is weak and requires more explanation and analysis.

Line 285: Figure 5 is rather poorly done.  The legend is completely lacking.  Please revise and follow standard conventions for writing a legend suitable for a published work.  I believe that the predicted promoter and expression cis elements for this family may be of interest, however this is weak – are any of these verified at the loci?  For example, in BoWHY1a there appear to be 14 light related elements identified.  How many of each have been genetically dissected and verified?  Are all of them accessible and functional at all times?  Often more complex chromatin structures can mask motifs and many may not be functional (e.g. it is easy to imagine a model where as a TF undergoes exon/intron loss that the need for expression may differ, resulting in erosion of some of the motifs).  Simply the identification of a DNA sequence on its own is of very little importance.  This would be best moved to the supplement or removed entirely if there is no dissection and verification of these.  Minor notes: using a heat map with the number in it (as in B) kind of diminishes the need for a heat map.  And the color scheme for 6 seems off from the red-crimson choices made by the authors for the rest of the figure.

Line 326: the authors should introduce the chloroplastic marker, which I am assuming is CHI on the figure that follows.  While many dyes and probes are well known and standard, such as DAPI, it is still standard to discuss what the authors are presenting to the reader.  I would suggest that they mention that DAPI is a DNA stain used to localize the nucleus, and they should introduce the CHI channel probe – this reviewer is not familiar with it. 

Line 338: In figure 8 I do not think that it is needed to Merge the DIC channel with the three fluorophores – it doesn’t add anything, and it makes the colocalization images that the authors are highlighting more challenging to clearly see.  In addition, the figure legend is written poorly, and it provides a two day window for when the green signal was obtained (either 2 or 3 days post transfection).  While the average reader will probably be able to assume that the DAPI channel and the CHI channel probably were imaged by laser confocal microscopy during the same timeframe, it would be beneficial to rewrite this section for clarity all the same.

Line 354: The RNA-seq dataset that was analyzed was accessed from the lab website… has this been peer reviewed and published?  Has it been deposited into a database (e.g. SRA)?  Are there GEO accession numbers, etc?  More information including accession numbers and references of the pubs would be needed for this work.

Line 388: Add more details into the figure legend.  The authors should specify that the expression is relative to Actin in the figure/legend.  The lettering is confusing – If the lettering indicates significant differences between data, how do the bars labelled a differ (left panel of part c – the first blue column is ab, the red is a, and the third is grey and has ab.  All of these are not significant if the SEM values are as large as presented, yet they are labelled as if there is a significant difference between each.  This needs to be clarified, corrected, and presented in a better manner.

Comments on the Quality of English Language

English is largely fine.  Minor editing required, but the authors should be quite capable of completing this themselves.

Author Response

(The authors gave the same response as above.)

Round 2

Reviewer 2 Report

Comments and Suggestions for Authors

The authors have successfully addressed all the concerns and revised the manuscript appropriately.